# Targeting C-Reactive Protein by Selective Apheresis in Humans: Pros and Cons

**DOI:** 10.3390/jcm11071771

**Published:** 2022-03-23

**Authors:** Jan Torzewski, Patrizia Brunner, Wolfgang Ries, Christoph D. Garlichs, Stefan Kayser, Franz Heigl, Ahmed Sheriff

**Affiliations:** 1Cardiovascular Center Oberallgaeu-Kempten, Clinic Association Allgaeu, 87439 Kempten, Germany; jan.torzewski@klinikverbund-allgaeu.de; 2Pentracor GmbH, 16761 Hennigsdorf, Germany; brunner@pentracor.de (P.B.); kayser@pentracor.de (S.K.); 3Medical Clinic, Diakonissenhospital Flensburg, 24939 Flensburg, Germany; rieswo@diako.de (W.R.); garlichsch@diako.de (C.D.G.); 4Medical Care Center Kempten-Allgaeu, 87439 Kempten, Germany; heigl@mvz-kempten.de; 5Division of Gastroenterology, Infectiology and Rheumatology, Medical Department, Charité University Medicine, 12200 Berlin, Germany

**Keywords:** inflammation, cardiovascular, COVID-19, arteriosclerosis, ischemic stroke, therapeutic apheresis

## Abstract

C-reactive protein (CRP), the prototype human acute phase protein, may be causally involved in various human diseases. As CRP has appeared much earlier in evolution than antibodies and nonetheless partly utilizes the same biological structures, it is likely that CRP has been the first antibody-like molecule in the evolution of the immune system. Like antibodies, CRP may cause autoimmune reactions in a variety of human pathologies. Consequently, therapeutic targeting of CRP may be of utmost interest in human medicine. Over the past two decades, however, pharmacological targeting of CRP has turned out to be extremely difficult. Currently, the easiest, most effective and clinically safest method to target CRP in humans may be the specific extracorporeal removal of CRP by selective apheresis. The latter has recently shown promising therapeutic effects, especially in acute myocardial infarction and COVID-19 pneumonia. This review summarizes the pros and cons of applying this novel technology to patients suffering from various diseases, with a focus on its use in cardiovascular medicine.

## 1. Introduction

In humans, C-reactive protein (CRP) activates the classical complement pathway via C1q [1] and stimulates macrophages via Fcγ-receptors [2,3]. Obviously, CRP utilizes the same biological structures as antibodies [4]. In further analogy, CRP may be causally involved in various human diseases by triggering (severe) ancient autoimmune reactions [5,6,7]. Although the latter hypothesis has been discussed in medical science since decades the issue has never been clarified. This is due to the fact that no drug or medical product targeting CRP has been on the market so far.

CRP synthesis and structure have extensively been reviewed elsewhere [8,9]. Here, we briefly review the role of CRP in physiology and pathophysiology with a focus on complement and macrophages. We then deal with the recent breakthrough in CRP targeting achieved by selective CRP apheresis. Pros and cons are listed in Table 1. Finally, we give an overview on current clinical trials and hypothesize on future developments. Thus, this review article may also be considered as an opinion paper.

## 2. CRP in Physiology

CRP is expressed in the ancient Limulus for more than 250 million years ago [10]. Although evolutionarily highly conserved, there are significant species differences in CRP function [11]. In humans, CRP activates the classical complement pathway and opsonizes biological particles for macrophages via Fcγ-receptors [1,2,3,12]. The latter seems remarkable as these functions are also antibody functions: Thus, like CRP, antibodies activate the classical complement cascade and bind to Fcγ-receptors via their Fc-region [4]. CRP has appeared earlier in evolution than antibodies and may, consequently, be the first antibody analogue in the evolution of the immune system [13]. In ancient *Limulus*, which survives without the benefits of adaptive immunity, CRP is vital for host defense against bacterial infection [10]. In humans, however, having developed highly sophisticated adaptive immunity, CRP may rather be a relic of evolution and emerged to have a role in tissue regeneration [6]. When apoptotic or dying cells display lysophosphatidylcholine (oxidized phosphatidylcholine) in their membranes, CRP recognizes these cells and opsonizes them for Fcγ-receptor mediated removal by macrophages [14]. Thus, in human physiology, CRP may, above all, play a major role in the process of wound healing and removal of apoptotic and necrotic cells.

CRP is probably the most commonly measured inflammatory molecule in clinical medicine. Again, as CRP activates complement via C1q and stimulates macrophages via Fcγ-receptors (in analogy to antibodies) CRP may be considered as an early primitive antibody and a pathogenic factor rather than an inflammation marker only.

## 3. CRP in Pathophysiology

In pathophysiology, an active contribution of CRP to initiation and progression of disease has been discussed for decades [13,15,16]. This discussion has never come to an end because a definitive proof of CRP’s causal involvement in disease in humans was lacking. Whereas the molecule’s role as a marker of activity of infectious, autoimmune, ischemic or even cardiovascular disease [17] is well established and generally accepted, it is important to note that there is no international consensus on causal contribution of CRP to the pathogenesis of any human disease. In cardiovascular disease, mendelian randomization trials strongly contradict causality and active contribution of CRP to pathogenesis [18,19,20]. This is crucial. It is, however, also crucial to realize that Mendelian randomization has to be interpreted with care and is, by far, less reliable than randomization in clinical trials. The latter has been reviewed in detail in a number of noteworthy articles [21,22,23,24]. Regulation of CRP synthesis includes not only one but a number of genes [25,26,27] and thus, a one gene/one protein genetic approach might be problematic. In addition, the issue of canalization could be relevant in a protein that is as highly conserved as CRP. Finally, cardiovascular disease is complex and each disease entity deserves detailed analysis [13]. In particular, the pathophysiological role of CRP in acute events must be considered separately from that in chronic events. The evidence in the acute setting is overwhelming (please see Section 3.3.2), whereas the evidence in the chronic inflammatory setting is still being collected. Ultimately, randomized trials might clarify these issues much better than Mendelian randomization [28]. Randomized controlled trials, however, are only possible with an available specific and efficient therapy comparing treatment group to control group. Such therapy has only very recently become reality.

### 3.1. CRP in Viral and Bacterial Infection

CRP is one of the most frequently determined molecules in clinical medicine. In daily practice, it is used for the non-specific initial diagnosis of viral or bacterial infection and also for monitoring the course of such infection under medical therapy [29]. Especially, the success of antibiotic therapy in bacterial infection and sepsis is usually determined by measuring CRP levels in addition to clinical evaluation. Importantly, CRP plasma levels in viral infection are usually significantly lower than CRP levels in bacterial infection [30]. This is of particular importance when looking at COVID-19 disease (please see Section 3.5). Although COVID-19 patients suffer from a viral disease, CRP levels in COVID-19 patients with a bad prognosis are surprisingly high. Plasma levels up to 400 mg/L, usually seen in severe bacterial infection or sepsis only, are common in deleterious COVID pneumonia without superinfection [31].

### 3.2. CRP in Autoimmune Disease

Although the association between CRP and the activity of autoimmune disease is well known and highly suggestive for a causal involvement in this heterogeneous group of diseases (like rheumatoid arthritis, ulcerative colitis, Crohn’s disease, psoriasis, giant cell arteritis etc.) [30], there are no studies on specific CRP inhibition in autoimmune disease. Notably, upstream interleukin-6 (IL-6) targeting with tocilizumab seems partly effective [32].

### 3.3. CRP in Cardiovascular Disease

#### 3.3.1. Atherosclerosis

CRP plasma levels correlate with cardiovascular risk [17]. CRP accumulates in human atherosclerotic lesions [33] and exerts pro-atherogenic effects in vitro [13]. Some of the effects reported in the literature, especially on endothelial cells and vascular smooth muscle cells, however, have been shown to be caused by contamination of the used CRP preparations by either azide or lipopolysaccharide [34,35]. The latter seems reasonable as CRP interaction with ancient immune cells, i.e., macrophages, is visible in tissue specimen and seems more relevant than interaction with cells not contributing to the immune response. Mendelian trials, in awareness of their limitations, contradict the significant causal contribution of CRP to atherogenesis and its sequelae [18,19,20]. In contrast, recent large clinical trials suggest that the IL-1β/IL-6/CRP pathway is intimately involved in cardiovascular disease [36,37,38]. Specific and direct targeting of the CRP molecule, however, has never been tried yet. The only reason for this is the fact that, in spite of huge pharmacological effort, no CRP specific chemical inhibitor or antagonist has been available.

#### 3.3.2. CRP in Myocardial Infarction

Acute myocardial infarction (AMI) implies a huge burden for the health system, since patients who recover still suffer from reduced quality of life and a high risk of severe complications later on. The risk correlates significantly with the extent of myocardial injury [39]. Especially innate immunity aggravates and extends myocardial injury [40,41]. Serum CRP concentrations during and after AMI correlate with clinical outcome and with larger infarct size [42,43,44,45]. This has been described for more than two decades now and is in line with the known pathological function of CRP: eliminating cells in the area at risk [6,46,47]. This area contains cells, which could recover after revascularization and reperfusion, but are sequentially destroyed by immune-mediated mechanisms. Numerous experimental approaches focusing specifically on AMI have shown this in detail [48,49,50,51,52,53]. Recently, it has been shown in the **C**-reactive protein apheresis in **A**cute **M**yocardial **I**nfarction-1 (CAMI-1) study that the magnitude of infarct damage or reduction in cardiac output is significantly related to the amount of CRP synthesized by the patient immediately after the onset of his AMI. In the same study, significant evidence was also found that reduction in CRP levels conferred a better outcome in terms of infarct size and cardiac output. Some patients in the verum group (CRP apheresis group) even showed no scar at all (as assessed by cardiovascular magnetic resonance). These were not aborted infarctions, because aborted infarctions were treated as dropouts [5].

#### 3.3.3. CRP in Myocarditis and Dilated Cardiomyopathy

In myocarditis, most frequently caused by viral infection [54], elevated CRP levels are common. Autoimmune myocarditis is also associated with high CRP plasma levels. Chronic myocarditis is known to trigger the development of dilated cardiomyopathy [54], a disease leading to ongoing heart failure not only in elder but also in younger patients. It is noteworthy that CRP and complement deposits have been shown to be frequently present in myocardial biopsy specimen obtained from patients suffering from dilated cardiomyopathy [55].

### 3.4. CRP in Neurological Disorders and Stroke

Ischemic stroke exhibits similar pathological mechanisms to AMI. To date, restoring rapid reperfusion of the brain constitutes the only established therapeutic strategy to reduce the size of the infarct and the consequences of the disease [56]. However, inflammation plays an important role in various stages of ischemic stroke. Several humoral and cellular mechanisms are set in motion by the occlusion and subsequent therapeutic reperfusion [57]. Several findings substantiate the hypothesis that CRP plays an identical pathological role as shown in AMI, facilitating the elimination of energetically challenged and compromised cells in the penumbra.

The early inflammatory response after stroke has been identified as a key prognostic factor [58,59]. Patients with favorable clinical outcome feature significantly lower levels of inflammatory parameters, especially CRP, compared to patients with poor outcome [60,61,62]. Muir et al. have shown that CRP levels measured within 72 h after stroke predict mortality over an observation period of up to 4 years [63]. Further, studies in a rat animal model have shown that infusion of human CRP enlarges cerebral infarct areas after acute occlusion via a complement-dependent mechanism [64].

### 3.5. CRP in COVID-19

COVID-19 is a virus-induced disease, but it also includes an important immune and autoimmune component. CRP was used early on during the pandemic as a marker for the severity and progression of the disease in patients, as it increases dramatically together with IL-6 during the clinical manifestation of COVID-19 [65,66,67,68]. The validity of CRP as a significant predictor of the outcome in COVID-19 was confirmed many times. A rapid increase in CRP allows the prognosis of ventilatory requirement of patients as well as their clinical outcome [31]. CRP levels further correlate with computed tomography (CT) findings in COVID-19 patients [69].

Corresponding to these findings, abundant amounts of CRP and complement deposits and were found in the lungs of deceased COVID-19 patients, including mainly C1q [70,71].

Severe progression occurs in roughly 14% of patients suffering from COVID-19 and in 5% this can lead to ventilator dependency with a serious prognosis [68,72,73,74]. An important therapeutic approach focuses on the treatment of acute respiratory failure—a major cause of mortality, followed by cardiac and septic complications. In the severe course of the disease, there is an initial cytokine storm, accompanied by a massive increase in the CRP concentration, followed by pulmonary fibrosis [75,76].

Intra-alveolar edema and hemorrhage are common observations in the lungs of COVID-19 patients, resulting in ischemic alveolar tissue. In COVID-19 pneumonia, there is massive damage to the alveoli as well as thrombus formation in the microcirculation. Complement binding to CRP leads to immigration of macrophages and, via increased expression of tissue factor, to thrombus formation. Both are exacerbated by high CRP levels and parallel to the underlying pathomechanism in other diseases, CRP causally enlarges destroyed tissue and contributes to irreversible tissue destruction [77].

These findings support the hypothesis stated early on that targeting CRP therapeutically can inhibit the lung deterioration and disease progression [6,7,78,79]. This innovative therapeutic approach for the early phase of severe COVID-19 is currently being used in three German hospitals. Three of the treated cases and one case series have already been published [7,80,81,82] and another publication on a case series has been submitted and can be viewed as a preprint (https://www.preprints.org/manuscript/202203.0029/v1; 16 March 2022). In the case series by Esposito et al., there is a marked improvement in COVID-19 pneumonia on imaging performed before and after CRP apheresis.

## 4. Why May CRP Apheresis Be a Breakthrough in CRP Targeting?

Over more than two decades, several approaches to target CRP have been discussed and tried by various researchers and pharmaceutical companies. These approaches include:CRP inhibition by antisense technologies [83];CRP inhibition by small molecular weight inhibitors [84];Inhibition of hepatic CRP synthesis [85];Inhibition of CRP mediated complement activation;Inhibition of CRP binding to its receptors.

Each of these approaches was well justified and was also partly promising. Interestingly, however, none of them resulted in a specific substance or medication that was applicable in human clinical practice. The latter seems remarkable because all the people and companies involved were well experienced in drug development. Several reasons were causal for the problems in generating specific CRP inhibitors: Antisense technology was, up to present day, not sufficiently effective. Small molecular weight inhibitors have not yet been transferred to human application. For inhibition of hepatic CRP synthesis, no specific novel substance was identified. Blockage of CRP-mediated complement activation and its C1q binding site turned out to be difficult for steric reasons. Inhibition of CRP binding to its receptors was also impossible because CRP receptors are also antibody receptors and thus, this approach may result in severe immunosuppression. Generally spoken, CRP—in its pentameric structure—is a molecule with structural redundancies difficult to interfere with. Its synthesis is complexly regulated involving different gene loci and furthermore, its highly dynamic regulation involving an up to 10,000-fold increase in plasma levels within few hours during acute phase response counteract an efficient synthesis inhibition or targeting.

The most important clinical concern about CRP targeting is the danger of immunosuppression with consecutive bacterial or viral infection and sepsis. Interestingly, in spite of significant reduction of CRP plasma levels via CRP apheresis in both disease entities, we have not observed such effects in our patients suffering from acute myocardial infarction or COVID-19 disease [5,7,80,81,82]. Thus, either CRP is not crucial in immune defense against microbial pathogens or the remaining CRP plasma levels after CRP apheresis are still sufficient. CRP apheresis is highly specific and does not relevantly influence other inflammatory markers or medication [86]. In the clinical setting, the most relevant apheresis procedure is lipid apheresis which mainly targets lipids and is far less specific [87,88].

Very often in clinical medicine simple approaches have turned out to be the best ones. Consequently, the idea to selectively remove CRP from the human plasma by specific and highly efficient extracorporeal apheresis [52] lacking severe side effects may finally turn out to be superior to other approaches. CRP apheresis may become beneficial in clinical medicine. This potential benefit, however, needs to be proven and fostered by an additional clinical trial program. The latter is currently ongoing.

## 5. Clinical Trials

“First in man”-application of CRP apheresis has been published in 2018 for a ST elevation myocardial infarction (STEMI) patient [89]. In this patient, post STEMI CRP plasma levels were lowered effectively and the patient experienced no side effects from CRP apheresis. The same was published for a small cohort of STEMI patients in 2019 [90]. The first and also the only clinical study on CRP apheresis in STEMI patients published up to the present day is the **C**-reactive protein apheresis in **A**cute **M**yocardial **I**nfarction-1 (CAMI-1) study [5]. CAMI-1 was a non-randomized multi-center pilot study which has investigated feasibility and safety of CRP apheresis. Although the clinical observations were promising and the significant correlation between post-infarction CRP amount and myocardial infarct size was significantly lost in the treatment group, CAMI-1 cannot be regarded as being conclusive because it was not randomized.

Clinical trials including apheresis present the issue of an adequate sham control and a double-blind design. Although a sham control is biostatistically speaking crucial in order to get valid results that are not biased by the placebo effect, including this in the apheresis procedure is ethically challenging. Few apheresis trials included adequate sham controls, one of the best examples being granulocyte/monocyte apheresis in chronic gut diseases [91,92]. Here, control patients were subjected to the same extracorporeal circuit, but blood was bypassed and did not pass the column. As patients and clinicians were blinded and only the conducting apheresis team knew which patients received the sham procedure these trials were considered double-blind. This could be a feasible design for future CRP apheresis trials. However, the underlying disease has to be taken into consideration. After STEMI, ischemic stroke and during COVID-19, patients are hospitalized and already in critical state. Submitting them to a 4–6 h extracorporeal circuit and sham procedure is ethically not justifiable. Hence, most apheresis trials do not include a sham control and have either historical controls or patients that receive standard therapy without apheresis [93].

Current clinical trials investigating the effect of CRP apheresis on the course of various human diseases are summarized on the Website of U.S. National Library of Medicine/ClinicalTrials.gov (https://www.clinicaltrials.gov/ct2/results?term=C-reactive+protein+apheresis, accessed on 16 March 2022).

Two of the studies attract special attention because, for their randomized, controlled, multi-center design, they can be considered as proof of principle trials. One of them is the “CRP Apheresis in STEMI”-trial (NCT04939805), initiated by University of Innsbruck, Austria, a randomized, multi-center interventional trial including 170 patients and comparing standard therapy of STEMI plus CRP apheresis to standard therapy alone. The study largely follows the CAMI-1 protocol, it is well-planned and well-organized. Whether it is adequately powered to finally detect statistically significant differences in a disease with heterogeneous underlying anatomy and pathology is a matter of concern. In this context, the CAMI-1 registry, a multi-center all-comer CRP apheresis in AMI registry may help to identify patient subgroups that profit best and may also help to plan another randomized trial with modified inclusion criteria.

The second study attracting special attention is the “CRP Apheresis for Attenuation of Pulmonary, Myocardial and/or Kidney Injury in COVID-19”-trial (NCT04898062), a randomized, controlled, multi-center interventional trial initiated by the University of Essen, Germany, including 50 patients and comparing standard therapy of COVID-19 plus CRP apheresis to standard therapy alone. This study is of considerable importance. It is based on the pathophysiological hypothesis that CRP, in COVID-19, triggers a fulminant innate immunity autoimmune reaction in the human body which may be the real cause for the deleterious course in subjects with severe COVID-19 (Please see Section 3.5). Like CAMI-1, it is based on published case reports and case series strongly suggesting that only few participants may power such a randomized study adequately in order to prove a therapeutic benefit of CRP apheresis in severe COVID-19 disease [7,80,82]. If so, this small trial might become a benchmark trial in demonstrating conclusively that CRP is a trigger of ischemia induced autoimmune responses in the human body.

Whether CRP apheresis is useful in atherosclerosis, myocarditis and dilated cardiomyopathy, neurological disorders and stroke, or even in autoimmune disease, requires further systematic investigation. As CRP apheresis is not yet a broadly established therapy, we propose to treat patients within a reputable scientific framework only, i.e., within a scientific registry or randomized, controlled trial.

## 6. Discussion and Future Developments

CRP has been the first antibody-like molecule in the evolution of the immune system. Surprisingly, although it appeared earlier in evolution than nowadays antibodies, CRP utilizes the same biological structures (C1q, FcγRs) and, by doing this, has similar functions as modern antibodies (activation of classical complement cascade, opsonization of biological particles for macrophages). Notably, like antibodies, CRP may cause autoimmune reactions in the human body. First, clinical evidence for this comes from a clinical pilot study on using C-reactive protein apheresis as an add on-treatment of myocardial infarction (CAMI-1) and from case reports on successful use of C-reactive protein apheresis in COVID-19 disease.

A definitive proof of principle, however, is still lacking, With the initiation of “CRP Apheresis in STEMI” and “CRP Apheresis for Attenuation of Pulmonary, Myocardial and/or Kidney Injury in COVID-19”, two randomized multicenter trials on C-reactive protein apheresis in STEMI on the one hand and COVID-19 on the other hand, a big step forward is to be expected soon after completion. These randomized trials are flanked by a number of clinical registries that may help to identify patient subgroups that strongly benefit from CRP apheresis. Finally, patients suffering from other acute and chronic diseases, in which CRP levels inversely correlate with prognosis (f. e. stroke, ulcerative colitis, Crohn’s disease, pancreatitis, chronic polyarthritis, atherosclerosis etc.) might benefit from CRP apheresis in the future.

## Figures and Tables

**Table 1 jcm-11-01771-t001:** Pros and cons for CRP apheresis.

Pros	Cons
efficient and fast removal of large amounts of CRP within hours	blood plasma needs to be supplied to the adsorber instead of whole blood
regenerable immune adsorber→nearly unlimited capacity	the treatment takes approximately 5 h and needs to be repeated on successive days depending on the indication
approved by CE certification for removal of CRP	additional anticoagulation maybe critical in some patients
specific for CRP	(minimally) invasive procedure requiring peripheral venous access or Shaldon catheter
no removal of other molecules or medication	immunosuppression (?)
reusable adsorber	
apheresis is an established technique

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
