# Peer review of "Targeting C-Reactive Protein by Selective Apheresis in Humans: Pros and Cons"

_jcm, 2022, doi:10.3390/jcm11071771_

Round 1

Reviewer 1 Report

The manuscript was reviewed the recent breakthrough in CRP targeting by selective CRP apheresis. 
This manuscript is well-written and the topic of the article is interesting and novel, which mentions
the CRP apheresis for the treatment of COVID 19. Some questions require further explanation by the 
authors.
Comments to the authors:
- According to the review, CRP apheresis appears to be a new therapy for patients with 
inflammation. However, there are many reasons for elevated CRP concentrations, is CRP 
apheresis effective for all inflammatory responses? Is CRP apheresis a treatment for a sign rather 
than root causes?
- Does CRP apheresis have side effects on the human body?
- Are there patients who are not suitable for CRP apheresis?
- There are many inflammatory markers, why target CRP? Is CRP apheresis the most efficient 
therapy than other inflammatory markers apheresis?
- It is suggested that the authors could create a Figure or Table about the advantages and 
disadvantages of the CRP apheresis compare to the standard therapy.
- In Line 256, it is suggested that the author could write the clinical trial number to let readers know 
about the current clinical trials of CRP apheresis

Author Response

Dear Reviewer No. 1,

thank you very much for your review which improves our manuscript considerably. Please find our comments below.

The manuscript was reviewed the recent breakthrough in CRP targeting by selective CRP apheresis.
This manuscript is well-written and the topic of the article is interesting and novel, which mentions
the CRP apheresis for the treatment of COVID 19. Some questions require further explanation by the
authors.

Thank you for this positive assessment.

Comments to the authors:
According to the review, CRP apheresis appears to be a new therapy for patients with inflammation. However, there are many reasons for elevated CRP concentrations, is CRP apheresis effective for all inflammatory responses? Is CRP apheresis a treatment for a sign rather than root causes?

Thank you for this important point. We do not know yet. There is evidence for acute myocardial infarction and COVID-19. Whether CRP apheresis is useful in atherosclerosis, myocarditis and dilated cardiomyopathy, neurological disorders and stroke, or even autoimmune disease, requires systematic investigation. We have now further clarified this point (page 8, lines 21-25).

- Does CRP apheresis have side effects on the human body?

Again, thank you for this important point. It should be mentioned that CRP apheresis can reduce the amount of CRP by a maximum of 84% which is the best performance observed so far. The most important clinical concern about CRP targeting in general is the danger of immunosuppression with consecutive bacterial or viral infection and sepsis. Interestingly, in spite of significant reduction of CRP plasma levels via CRP apheresis in both disease entities, we have not observed such effects in our patients suffering from acute myocardial infarction or COVID-19 disease. We have now further clarified this point (page 6 f., lines 37-45)

- Are there patients who are not suitable for CRP apheresis?

Thank you. As CRP apheresis is not yet a broadly established therapy, we propose to treat patients within a reputable scientific framework only, i.e. within a scientific registry or randomized trial. We have now further clarified this point as per your suggestion (page 8, lines 21-25)

- There are many inflammatory markers, why target CRP?

Thank you again. CRP is probably the most commonly measured inflammatory molecule in clinical medicine. As CRP activates complement via C1q and stimulates macrophages via Fcg-receptors (in analogy to antibodies) we consider CRP as an early primitive antibody and a pathogenic factor rather than an inflammation marker only. We have now further clarified this point as per your suggestion (page 2, lines 35-38)

Is CRP apheresis the most efficient therapy than other inflammatory markers apheresis?

CRP apheresis is highly specific and does not relevantly influence other inflammatory markers. In the clinical setting, the most relevant apheresis procedure is lipid apheresis which mainly targets lipids and is far less specific. We have also further clarified this issue now (page 6 f., lines 37-45).

- It is suggested that the authors could create a Figure or Table about the advantages and
disadvantages of the CRP apheresis compare to the standard therapy.

As per your suggestion (table 1).

- In Line 256, it is suggested that the author could write the clinical trial number to let readers know
about the current clinical trials of CRP apheresis.

As per your suggestion (page 7, line 43 and page 8, line 8).

In addition, we have updated the COVID-19 part of the review (paragraph 3.5).

Reviewer 2 Report

General comments: This study highlights the potential of CRP apheresis in treating diseases such as AMI, stroke and most importantly may reduce the severity of COVID-19. The content of this paper is accepted for publication. However, it requires improvisation in terms of sentence construction with grammatical correction throughout this manuscript. Kindly spell out the abbreviations used in this manuscript prior to using them throughout this manuscript. I noticed there is an abbreviation that comes later in the conclusion section. 

However, while reading this manuscript, I have yet to figure out clearly the cons part of the CRP apheresis review throughout this manuscript. Perhaps can highlight for each section, each of the pro and cons parts of CRP apheresis.

Specific comments: 

Page 2 line 44- suggesting to improve this sentence to CRP is expressed in the ancient Limulus for more than 250 million years ago. 

Page 2, line 48, 51,59- a huge gap between Fc and receptor

Page 2, line 70, 71, 72: Suggesting to improve this sentence "The major reason for this is that, in cardiovascular disease (which, due to the high number of patients, attracts significant attention), mendelian randomization trials strongly contradict causality and active contribution to pathogenesis (Zacho et al., 2008; Elliott et al., 2009; Wensley et al., 2011). This is crucial."

Page 3 Line 119, 120: Mendelian trials, in awareness of their limitations, contradict the significant causal contribution of CRP to atherogenesis and its sequelae (please see above) - Please see above is referring to a figure?. If not, suggesting to reconstructing this sentence to make it more reader-friendly. 

Page 3, lines 122, 123: "Specific direct CRP targeting, due to its lack of availability, has never been tried yet". This sentence is unclear either there is an absence of clinical trials that targetted CRP as a clinical outcome (which is unlikely) or the results from the clinical trial showed a non-beneficial reduction in CRP?

Page 3 lines 125,129: Suggesting reconstructing the sentence to make it more reader-friendly with the correct grammar. 

Page 3 lines 142-144: This sentence needs further elaboration. What are the Verum group and the scar is referring to? 

 Page 4: Line 179: CT is referring to?

Page 5: Lines 215-217, 219-222, 227-233: These sentences require revision. 

- Suggesting adding a paragraph on which clinical trials failed to show good outcomes in terms of CRP reduction thus leading to the importance of  CRP apheresis rather than specific drug treatment. Maybe in CRP in Pathophysiology section (3). 

Author Response

Dear Reviewer No. 2,

thank you very much for your review which improves our manuscript considerably. Please find our comments below.

General comments: This study highlights the potential of CRP apheresis in treating diseases such as AMI, stroke and most importantly may reduce the severity of COVID-19. The content of this paper is accepted for publication. However, it requires improvisation in terms of sentence construction with grammatical correction throughout this manuscript.

Thank you for this positive assessment.

Kindly spell out the abbreviations used in this manuscript prior to using them throughout this manuscript. I noticed there is an abbreviation that comes later in the conclusion section.

As per your suggestion (page 4, line 32).

However, while reading this manuscript, I have yet to figure out clearly the cons part of the CRP apheresis review throughout this manuscript. Perhaps can highlight for each section, each of the pro and cons parts of CRP apheresis.

In accordance with both reviewers` suggestions we have now included a table summarizing the pros and cons of CRP apheresis (please see table 1).

Specific comments:
Page 2 line 44- suggesting to improve this sentence to CRP is expressed in the ancient Limulus for more than 250 million years ago.

As per your suggestion (page 2, line 17).

Page 2, line 48, 51,59- a huge gap between Fc and receptor

As per your suggestion (page 2, lines 20, 23 and 32).

Page 2, line 70, 71, 72: Suggesting to improve this sentence "The major reason for this is that, in
cardiovascular disease (which, due to the high number of patients, attracts significant attention), mendelian randomization trials strongly contradict causality and active contribution to pathogenesis (Zacho et al., 2008; Elliott et al., 2009; Wensley et al., 2011). This is crucial."

As per your suggestion (page 3, lines 2-3).

Page 3 Line 119, 120: Mendelian trials, in awareness of their limitations, contradict the significant causal contribution of CRP to atherogenesis and its sequelae (please see above) - Please see above is referring to a figure? If not, suggesting to reconstructing this sentence to make it more reader-friendly.

As per your suggestion (page 4, line 12).

Page 3, lines 122, 123: "Specific direct CRP targeting, due to its lack of availability, has never been tried yet". This sentence is unclear either there is an absence of clinical trials that targeted CRP as a clinical outcome (which is unlikely) or the results from the clinical trial showed a non-beneficial reduction in CRP?

As per your suggestion (page 4, lines 14-17).

Page 3 lines 125,129: Suggesting reconstructing the sentence to make it more reader-friendly with the correct grammar.

As per your suggestion (page 4, lines 14-17).

Page 3 lines 142-144: This sentence needs further elaboration. What are the Verum group and the scar is referring to?

As per your suggestion (page 4, lines 37-38).

Page 4: Line 179: CT is referring to?

As per your suggestion (page 5, line 28).

Page 5: Lines 215-217, 219-222, 227-233: These sentences require revision.
- Suggesting adding a paragraph on which clinical trials failed to show good outcomes in terms of CRP reduction thus leading to the importance of CRP apheresis rather than specific drug treatment. Maybe in CRP in Pathophysiology section (3).

As per your suggestion (page 6, lines 22-32). We would like to emphasize here that, up to the present day, there is no specific CRP inhibitor for human application. Other studies have used antibodies to Il-1ß or IL-6, the major CRP inducing cytokines. These antibodies, however, are far less specific than CRP apheresis because IL1-ß and IL-6 (as commonly known) also inhibit pathways other than just the IL-1ß/IL-6/CRP axis. This is now further specified (page 6, lines 22-32).

In addition, we have updated the COVID-19 part of the review (paragraph 3.5).

Round 2

Reviewer 1 Report

 Accept in present form.